# ELASTOGEN: 4D GENERATIVE ELASTODYNAMICS

## ABSTRACT

We present ElastoGen, a *knowledge-driven* AI model that generates physically accurate 4D elastodynamics. Unlike deep models that learn from video- or image-based observations, ElastoGen leverages the principles of physics and learns from established mathematical and optimization procedures. The core idea of ElastoGen is converting the differential equation, corresponding to the nonlinear force equilibrium, into a series of iterative local convolution-like operations, which naturally fit deep architectures. We carefully build our network module following this overarching design philosophy. ElastoGen is much more lightweight in terms of both training requirements and network scale than deep generative models. Because of its alignment with actual physical procedures, ElastoGen efficiently generates accurate dynamics for a wide range of hyperelastic materials and can be easily integrated with upstream and downstream deep modules to enable end-to-end 4D generation.

## 1 INTRODUCTION

Recent advancements in generative models have enhanced the ability to produce high-quality digital contents across diverse media formats (e.g. images, videos, 3D models, 4D data). In particular, the generation of 4D data, including both spatial and temporal dimensions, has seen notable progress (Singer et al., 2023; Shen et al., 2023; Xu et al., 2024; Ling et al., 2023; Bahmani et al., 2024a; Yin et al., 2023; Bahmani et al., 2024b).

On the other hand, learning physical dynamics that exhibit temporal consistency and adhere to physical laws from observable data remains a difficult problem. Data are in the wild and noisy. Their underlying coherence is agnostic to the user. As a result, existing deep models have to assume some distributions of the data, which may not be the case in reality. In theory, the network would extract any knowledge provided sufficient data. In practice however, such data-based learning becomes more and more cumbersome with increased dimensionality of generated contents – it is unintuitive to define the right network structure to guide a physically meaningful generation; it requires terabyte- or petabyte-scale high-quality training data, and center-level computing resource to facilitate the training. Those theoretical and practical obstacles combined impose significant challenges.

We explore a new way to establish physics-in-the-loop generative models. Our argument is that *learning from knowledge* instead of from raw data is more effective for generative models. Physical laws and principles are often in the form of partial differential equations (PDEs) and numerically solved with discretized differential operators. We note that those operators hold a similar structure as a convolution kernel on the problem domain, where the values of those convolution kernels depend on the specific problem setting. Inspired by those observations, we propose ElastoGen, a knowledge-driven neural model that generates physically accurate and coherent 4D elastodynamics. ElasoGen can be easily coupled and integrated with upstream and downstream neural modules to enable end-to-end 4D generation. The core idea of ElastoGen is converting the global differential operator, corresponding to the nonlinear force equilibrium, to iterative local convolution-like procedures. Such knowledge-level priors allow us to design dedicated network modules for ElastoGen, where each network module has a well-defined purpose of relaxing locally concentrated strain rather than being treated as a piece of a "black box". Compared with other data-learning-based generative models, ElasoGen is lightweight – in terms of both training requirements and the network scales. Furthermore, due to its consistency with physics procedure, ElastoGen generates physically accurate dynamics for a wide range of hyperelastic materials. Specifically, we summarize some features of ElastoGen as follows:

**Compact generative network inspired by physics principles**   The network architecture of Elasto-Gen is strongly inspired by our prior knowledge of physics and corresponding numerical procedures. This allows a compact and effective generative framework in the form of deep neural networks. The training efforts for such a carefully tailored deep model become lightweight as well.

**NeuralMTL with diffusion parameterization**   ElastoGen features a so-called *NeuralMTL* module to encode the underlying constitutive relations for real-world hyperelastic materials such as Neo-Hookean and or Saint Venant-Kirchhoff (StVK). We leverage a lightweight conditional diffusion model to predict its network parameters to isolate our training efforts.

**Nested RNN with low-frequency encoding**   ElastoGen constitutes a two-level RNN architecture. We augment ElastoGen with a low-frequency encoder, which extracts low-frequency dynamic signals so that the local relaxation only takes care of the remaining high-frequency strains. This design makes ElastoGen more efficient for stiff instances.

## 2   RELATED WORK

**Generative models**  The primary objective of generative models is to produce new, high-quality samples from vast datasets. These models are designed to learn and understand the distribution of data, thereby generating samples that meet specific criteria. Techniques such as Generative Adversarial Networks (GANs) (Goodfellow et al., 2014), Variational Autoencoders (VAEs) (Kingma & Welling, 2014), and flow-based methods (Dinh et al., 2015; 2017) have all demonstrated significant success. However, each method has its limitations. For instance, GANs can generate high-quality images but are notoriously difficult to train and optimize (Arjovsky et al., 2017; Gulrajani et al., 2017; Mescheder, 2018). VAEs (Vahdat & Kautz, 2020; Child, 2021) and flow-based methods (Kingma & Dhariwal, 2018) offer efficient training processes but generally fall short in sample quality compared to GANs. Recently, diffusion models have emerged as another powerful technique, achieving state-of-the-art results in generating high-fidelity images (Sohl-Dickstein et al., 2015; Ho et al., 2020; Rombach et al., 2022), setting the stage for further explorations in more complex applications.

**4D generation based on diffusion models**  As research on diffusion models advances, these methods could potentially be applied to the generation of 3D content (Jain et al., 2022; Lin et al., 2023; Metzer et al., 2023; Poole et al., 2022; Wang et al., 2024b; Liu et al., 2023; 2024), video content (Blattmann et al., 2023; Harvey et al., 2022; Ho et al., 2022b;a; Karras et al., 2023; Ni et al., 2023), and more complex forms such as 3D videos or what might be termed 4D scenes (Singer et al., 2023; Shen et al., 2023; Xu et al., 2024; Ling et al., 2023; Bahmani et al., 2024a; Yin et al., 2023; Bahmani et al., 2024b). These advanced applications demonstrate the versatility and expanding potential of diffusion models across diverse media formats. However, existing video generation techniques struggle to ensure temporal consistency and require substantial training data, underscoring the challenges of capturing and replicating the dynamic and interconnected behaviors present in real-world scenarios within a generative model framework.

**Neural physical synamics**  Physical dynamics traditionally relies on numerical solutions such as the finite element method (FEM) (Zienkiewicz & Morice, 1971; Zienkiewicz et al., 2005; Huebner et al., 2001; Reddy, 1993), finite difference method (Zhu et al., 2010; Godunov & Bohachevsky, 1959), or mass-spring systems (Liu et al., 2013). Each approach offers distinct advantages and limitations. For example, Position-Based Dynamics (PBD) (Müller et al., 2007) and Projective Dynamics (PD) (Bouaziz et al., 2014; Liu et al., 2013) offer simplified implementation and faster convergence but can struggle with complex material behaviors and do not always guarantee consistent convergence rates. Recently, neural physics solvers, which integrate neural networks with traditional solvers, aim to accelerate and simplify the computation process. The pioneering works (Chang et al., 2017; Battaglia et al., 2016) directly utilized neural networks to predict dynamics, achieving promising results in simple particle systems. Subsequent studies (Sanchez-Gonzalez et al., 2018; Kipf et al., 2018; Ajay et al., 2018; Li et al., 2019c;a;b) adopted network architectures to the specific features of the systems, thereby enhancing performance. The advent of Physics Informed Neural Networks (PINNs) (Raissi et al., 2019; Pakravan et al., 2021) marks a leap forward. These networks incorporate extensive physical information to constrain and guide the learning process, ensuring that predictions adhere more closely to physical laws and has succeeded in domains such as cloths (Geng et al., 2020) and fluids (Um et al., 2020; Gibou et al., 2019; Chu et al., 2022). Some work (Yang et al., 2020) shifts away from end-to-end structures and use neural networks to optimize part of the

simulation. Another line of research generates dynamics through physics-based simulators, where network learns static information while physical laws govern the generation of dynamics (Li et al., 2023; Feng et al., 2023; Xie et al., 2023; Feng et al., 2024; Jiang et al., 2024), giving physical meanings to Neural Radiance Fields (NeRF) (Mildenhall et al., 2020; Kerbl et al., 2023a). These methods demonstrate the benefits of embedding human knowledge into networks to reduce the learning burden.

## 3 BACKGROUND

To make the paper more self-contained, we start with a brief review of some preliminaries of a dynamic elastic model.

### 3.1 VARIATIONAL OPTIMIZATION OF ELASTODYNAMICS

The dynamic equilibrium of a 3D model can be characterized by $\frac{\mathrm{d}}{\mathrm{d}t}\left(\frac{\partial L}{\partial \dot{\mathbf{q}}}\right) - \frac{\partial L}{\partial \mathbf{q}} = \mathbf{f}_q$, where $L = T - U$ is system *Lagrangian* i.e., the difference between the kinematic energy ($T$) and the potential energy ($U$). $\mathbf{q}$ and $\dot{\mathbf{q}}$ are generalized coordinate and velocity. $\mathbf{f}_q$ is the generalized external force. With the implicit Euler time integration scheme: $\mathbf{q}_{n+1} = \mathbf{q}_n + h\dot{\mathbf{q}}_{n+1}$, $\dot{\mathbf{q}}_{n+1} = \dot{\mathbf{q}}_n + h\ddot{\mathbf{q}}_{n+1}$, it can be reformulated as a nonlinear optimization to be solved at each time step:

$$\mathbf{q}_{n+1} = \underset{\mathbf{q}}{\mathrm{argmin}} \left\{ \frac{1}{2h^2}\|\mathbf{q} - \mathbf{q}_n - h\dot{\mathbf{q}}_n - h^2\mathbf{M}^{-1}\mathbf{f}_q\|_{\mathbf{M}}^2 + U(\mathbf{q}) \right\}, \tag{1}$$

where the subscript indicates the time step index. $h$ is the time step size, and $\mathbf{M}$ is the mass matrix.

### 3.2 DIFFUSION MODEL

A diffusion model transforms a probability from the real data distribution $\mathcal{P}_{\text{real}}$ to a target distribution $\mathcal{P}_{\text{target}}$ through diffusion and denoising.

**Diffusion.** The diffusion process incrementally adds Gaussian noise to the initial data $\mathbf{x}_0 \sim \mathcal{P}_{\text{target}}$, gradually transforming it into a sequence $\mathbf{x}_1, \mathbf{x}_2, ..., \mathbf{x}_T$, where $\mathbf{x}_T$ approximates the real distribution $\mathcal{P}_{\text{real}}$. The aim is to learn a noise prediction model $\boldsymbol{\epsilon}_\theta(\mathbf{x}_t, t)$, estimating the noise at each iteration $t$ to facilitate data recovery in the denoising phase. The noise learning objective is formulated as:

$$L = \mathbb{E}_{\mathbf{x}_0 \sim \mathcal{P}_{\text{target}}, \boldsymbol{\epsilon} \sim \mathcal{N}(\mathbf{0}, \mathbf{I}), t \sim \text{Uniform}(\{1, ..., T\})} [\|\boldsymbol{\epsilon} - \boldsymbol{\epsilon}_\theta(\mathbf{x}_t, t)\|^2]. \tag{2}$$

**Denoising.** Denoising iteratively removes noise from $\mathbf{x}_T \sim \mathcal{P}_{\text{real}}$, recovering the original data $\mathbf{x}_0$ by adjusting the noisy data at each iteration $t$ as:

$$\mathbf{x}_{t-1} = \frac{1}{\sqrt{\alpha_t}}(\mathbf{x}_t - \frac{1 - \alpha_t}{\sqrt{(1 - \overline{\alpha_t})}}\boldsymbol{\epsilon}_\theta(\mathbf{x}_t, t)) + \sigma_t \mathbf{z}, \quad \mathbf{z} \sim N(\mathbf{0}, \mathbf{I}), \tag{3}$$

where $1 - \alpha_t = \beta_t$ is a scheduled variance, and $\sigma_t$ is typically set to $\sigma_t = \sqrt{\beta_t}$. $N(\mathbf{0}, \mathbf{I})$ is standard normal distribution. Diffusion and denoising processes allow for effective modeling of the transition between distributions, using learned Gaussian transitions for noise prediction and reduction.

## 4 METHODOLOGY

Our overall pipeline is visualized in figure 1. ElastoGen is a lightweight generative deep model producing physically grounded 4D contents given some general descriptions of the object e.g., stiffness or density.Such information could also be learned via observations since ElastoGen is trivially differentiable. ElastoGen rasterizes an input shape and leverages a nested two-level RNN to predict its further trajectory sequentially. Each prediction is subject to an accuracy check to ensure the result is physically accurate. Such network structure adheres to a well-reasoned numerical procedure for solving the variational optimization of equation 1. Therefore, ElastoGen does not have redundant or purpose-less network components that could potentially lead to overfitting. In the following sections, we elaborate on each major module of our pipeline.

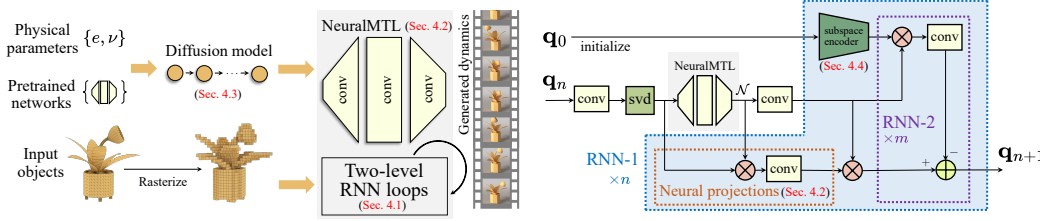

(a) The pipeline of ElastoGen.                    (b) The network structure.

Figure 1: **Pipeline overview. (a)** ElastoGen rasterizes an input 3D model (with boundary conditions) and generates parameters filling our NeuralMTL module. Conceptually, NeuralMTL predicts locally concentrated strain of the object, which is relaxed by a nested RNN loop. **(b)** The RNN predicts the future trajectory of the object. There are two sub RNN modules. RNN-1 repeatedly relaxes the local stress in a 3D convolution manner. Those relaxed strains are converted to positional signals, and RNN-2 merges local deformation into a displacement field of the object. ElastoGen automatically checks the accuracy of the prediction of both RNN loops, and outputs the final prediction of $\mathbf{q}_{n+1}$ once the prediction error reaches the prescribed threshold.

## 4.1 METHOD OVERVIEW: PIECE-WISE LOCAL QUADRATIC APPROXIMATION

Our elastodynamic generation mimics numerical optimization procedures that minimize the variational energy of equation 1. It is possible to tackle this problem at the global level, i.e., optimizing all the degrees of freedom (DoFs) of the system at once e.g., using Newton's method. Such a brute-force scheme requires to learn dense inter-correlations among features at all DoFs, which inevitably leads to complex and large-scale network architectures with numerous parameters to be learned.

Alternatively, we opt for a divide-and-conquer way to approach equation 1. We consider the total potential energy $U$ as the summation of multiple energies of quadratic form: $U(\mathbf{q}) \approx \sum_i E_i(\mathbf{q}_i)$, where $E_i(\mathbf{q}) = \min_{\mathbf{p}_i \in \mathcal{M}_i} \frac{\omega_i}{2} \|\mathcal{G}_i[\mathbf{q}_i] - \mathbf{p}_i\|^2$. Here, $i$ indicates the $i$-th sub-volume of the object. For instance, one may discretize the object into a tetrahedral mesh, and $E_i$ then represents the elastic potential stored at the $i$-th element. $\mathcal{G}_i$ denotes a *discrete differential operator*, which converts positional features $\mathbf{q}_i$ to strain-level features. To this end, we build $\mathcal{G}_i$ such that $\mathcal{G}_i[\mathbf{q}_i] = \text{vec}(\mathbf{F}_i)$, i.e., the vectorized deformation gradient ($\mathbf{F} \in \mathbb{R}^{3 \times 3}$) of the sub-volume, which gives the local first-order approximation of the displacement field. The constraint manifold $\mathcal{M}_i$ denotes the zero level set of $E_i$. In other words, we consider $E_i$ as a quadratic energy based on how far local displacement $\mathbf{q}_i$ is from its closest energy-free configuration ($\mathbf{p}_i$), given the local material stiffness $\omega_i$.

Provided the current deformed shape $\mathbf{q}_i$, we can find $\arg \min_{\mathbf{p}_i} \frac{\omega_i}{2} \|\mathcal{G}_i[\mathbf{q}_i] - \mathbf{p}_i\|^2$, which suggests a locally optimal descent direction to reduce $U$. The global displacement can then be obtained by minimizing $\mathbf{q}$ over $E_i$ at all the sub-volumes. While this is a global operation that we would like to avoid, it is essentially a Laplacian-like smoothing operator, which can still be processed with repeated local smoothing. This procedure share a similar nature of shape matching method (Müller et al., 2005) and PD (Bouaziz et al., 2014) — it offers a piece-wise SQP way (Boggs & Tolle, 1995) to approximate $U$ locally. ElastoGen functions like a neural version of the aforementioned procedure with a nested RNN structure. It handles local solve, or strain *relaxation* in the form of a volume convolution so that the overall network structure is compact and lightweight.

Unfortunately, real-world materials are more than a collection of quadratic forms. The appropriate $\mathcal{M}_i$ nonlinearly vary under different deformation or material models, aka material nonlinearity. As a result, shape matching or PD can only handle simplified material behavior unless we know how $\mathcal{M}_i$ changes along the generation. To this end, we augment ElastoGen with a NeuralMTL module to make sure each local SQP matches actual materials.

## 4.2 NEURALMTL & NEURAL PROJECTION

The goal of NeuralMTL is to correct local quadratic approximations of $U$ so that ElastoGen faithfully generates physically accurate results for any real-world hyperelastic material. Specifically with

NerualMTL ($\mathcal{N}$), $E_i$ becomes:

$$E_i(\mathbf{q}_i) = \underset{\mathbf{P}_i \in \mathcal{SO}(3)}{\operatorname{argmin}} \frac{\omega_i}{2} \left\| \mathbf{F}_i \cdot \mathcal{N}\big(\mathcal{G}_i[\mathbf{q}_i]\big) - \mathbf{P}_i \right\|_F^2. \tag{4}$$

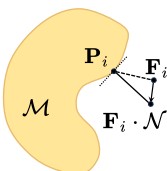

We set $\omega_i$ as $\omega_i = V_i e/(2(1+\nu))$ based on real-world material parameters: Young's modulus $e$, Poisson's ratio $\nu$ as well as the size of the sub-volume $V_i$. $\mathcal{G}_i$ extracts the deformation gradient $\mathrm{vec}(\mathbf{F}_i)$ and feeds it to NeuralMTL, $\mathcal{N}$. As the name suggests, $\mathcal{N}$ predicts a *neural strain* based on the information of local deformation $\mathbf{F}_i$. Given the material model and parameters, $\mathcal{N}$ is used for all $E_i$, and we do not put a subscript on $\mathcal{N}$. $\|\cdot\|_F$ denotes the Frobenius norm. $\mathcal{N}$ predicts a material-space strain prediction, which is then converted to is world space by $\mathbf{F}_i$. $\mathbf{P}_i \in \mathbb{R}^{3\times 3}$ is a rotation matrix i.e., $\mathbf{P}_i \in \mathcal{SO}(3)$. Intuitively, NeuralMTL warps $\mathbf{F}_i$ to a different configuration of $\mathbf{F}_i \cdot \mathcal{N}(\mathbf{F}_i)$ so that the new distance to $\mathbf{P}_i$ correctly reflects the local energy landscape of $E_i$ as visualized in the left inset.

For isotropic elastic materials, we add a nonlinear SVD (singular value decomposition) activation to the operator $\mathcal{G}_i$ such that $\mathbf{F}_i = \mathbf{U}_I \mathbf{S}_I \mathbf{V}_i^\top$. $\mathbf{S}_i$ is a diagonal matrix with singular values arranged in descending order, which correspond to the local principal strains. This activation converts $E_i$ to:

$$\begin{aligned} E_i(\mathbf{q}_i) &= \frac{\omega_i}{2} \| \mathbf{U}_i \mathbf{S}_i \mathbf{V}_i^\top \cdot \mathcal{N}(\mathcal{G}_i[\mathbf{q}_i]) - \mathbf{U}_i \mathbf{V}_i^\top \|_F^2 \\ &= \frac{\omega_i}{2} \mathrm{tr} \left( \mathbf{S}_i \mathbf{S}_i \mathbf{V}_i^\top \cdot \mathcal{N}(\mathcal{G}_i[\mathbf{q}_i]) \cdot \mathcal{N}^\top(\mathcal{G}_i[\mathbf{q}_i]) \mathbf{V}_i + \mathbf{I} - 2\mathbf{V}_i \mathbf{S}_i \mathbf{V}_i^\top \cdot \mathcal{N}(\mathcal{G}_i[\mathbf{q}_i]) \right). \end{aligned} \tag{5}$$

We further require this learning-based strain measure that 1) NeuralMTL predicts a symmetric strain; and 2) the adjusted energy remains invariant to rotation and merely depends on $\mathbf{S}_i$. Let $\mathbf{N}_i = \mathcal{N}(\mathcal{G}_i[\mathbf{q}_i]) \in \mathbb{R}^{3\times 3}$ be the raw output of NeuralMTL. Instead of directly imposing those restrictions during the training, we append a network module to nonlinearly activate the raw output of $\mathcal{N}$ as:

$$\mathcal{N}(\mathcal{G}_i[\mathbf{q}_i]) \leftarrow \mathbf{V}_i \big( \mathbf{N}_i + \mathbf{N}_i^\top \big) \mathbf{V}_i^\top, \tag{6}$$

which further simplifies $E_i$ to:

$$E_i = \frac{\omega_i}{2} \mathrm{tr} \left( \mathbf{Q}_i \mathbf{Q}_i^\top \right) + \frac{3\omega_i}{2} - \omega_i \, \mathrm{tr} \left( \mathbf{Q}_i \right), \quad \mathbf{Q}_i(\mathbf{S}_i) = \mathbf{S}_i \left( \mathbf{N}_i + \mathbf{N}_i^\top \right). \tag{7}$$

Intuitively, this activation escalates the order of the neural strain predicted by $\mathcal{N}$, pushing it to become a nonlinear strain estimation with a prescribed format — just like upgrading an infinitesimal strain to Green's strain to better measure large rotational deformation. It should be noted that NerualMTL prediction not alter the location of $\mathbf{P}_i$. As a result, the neural projection corresponding to our NerualMTL can be easily obtained as $\mathbf{P}_i = \mathbf{U}_i \mathbf{V}_i^\top$, i.e., the rotational component from $\mathbf{F}_i$. This is an important property of NerualMTL — if we choose to employ the network to learn an adjustment of $\mathbf{P}_i$ (which is also technically feasible), the local relaxation that predicts $\mathbf{P}_i$ becomes complicated, and the generation is less robust.

Given an input 3D object, ElastoGen rasterizes it into a set of 3D voxels. For a user-specified sub-volume e.g., in our implementation, each sub-volume is a voxel that intersects with the object, $\mathcal{G}_i$ operator extracts the local covariance matrix of the displacement field over this sub-volume. Let $\mathbf{A}_i = [\mathbf{q}_1, \mathbf{q}_2, ...\mathbf{q}_k] \in \mathbb{R}^{3\times k}$ and $\bar{\mathbf{A}}_i = [\bar{\mathbf{q}}_1, \bar{\mathbf{q}}_2, ...\bar{\mathbf{q}}_k]$ be deformed and rest-shape position of vertices of a sub-volume with $k$ vertices ($k = 8$ for a cubic volume). $\mathcal{G}_i$ has an analytic format of:

$$\mathcal{G}_i[\mathbf{q}_i] = \left[ \left( \bar{\mathbf{A}} \bar{\mathbf{A}}^\top \right)^{-1} \bar{\mathbf{A}} \otimes \mathbf{I} \right] \mathbf{q}_i, \tag{8}$$

which is an MLP whose weights can be pre-computed given the rasterized object. The output of $\mathcal{G}_i$ is then activated via a SVD module, which outputs $\mathbf{U}_i$, $\mathbf{V}_i$, and $\mathbf{S}_i$. As mentioned, $\mathbf{U}_i \mathbf{V}_i^\top$ constitutes the output of the local neural projection, but the energy check is performed through $\mathcal{N}$, which is embodied as a per-voxel compact convolution neural net. The weight coefficients of NeuralMTL are predicted by a generative diffusion model given the material type and parameters such as Young's modulus $e$ and Poisson's ratio $\nu$.

### 4.3 DECOUPLE NEURALMTL FROM MATERIAL PARAMETERS

As mentioned, NeuralMTL takes input as $\mathbf{F}_i$ and outputs $\mathcal{N}(\mathbf{F}_i)$, a neural strain measure. This learned strain is then fit to equation 4 to check if ElastoGen reasonably minimizes equation 1 and

is ready for the next time step. NeuralMTL is expected to fully accommodate material nonlinearity. Therefore, different material parameters $\{e, \nu\}$ guide NeuralMTL to yield different outputs even under the same $\mathbf{F}_i$. A straightforward approach is to train NeuralMTL $\mathcal{N}(\mathbf{F}_i, e, \nu)$ directly on both $\mathbf{F}_i$ and $\{e, \nu\}$. However, as NeuralMTL needs to be evaluated more frequently under different $\mathbf{F}_i$ (during the deformation) after the material parameters are given, we decouple the influences of $\mathbf{F}_i$ and $\{e, \nu\}$ to keep the network even more compact. Inspired by Zhang et al. (2024a), we note that NeuralMTL $\mathcal{N}(\mathbf{F}_i)$ can be generated using another diffusion network guided by $\{e, \nu\}$ such that $\mathbf{W} = \mathcal{D}(e, \nu)$, where $\mathbf{W}$ is the parameters of the network $\mathcal{N}(\mathbf{F}_i)$ and $\mathcal{D}$ is another diffusion model.

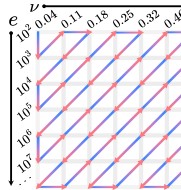

To train the model $\mathcal{D}$, we prepare a dataset of paired $\{e, \nu\}$ and $\mathbf{W}$. To this end, we first uniformly sample both $e$ and $\nu$ at fixed intervals and then establish a topological order, as shown in the left inset. A target elastic energy $\Psi(e, \nu)$ can be easily computed for each sampled $\{e, \nu\}$. $\mathbf{W}$ is then obtained via the following optimization:

$$\mathbf{W} = \underset{\mathbf{W}}{\operatorname{argmin}} \left\| \log(\frac{\omega_i}{2} \|\mathbf{F}_i \cdot \mathcal{N}(\mathbf{W}, \mathbf{F}_i) - \mathbf{U}_i \mathbf{V}_i^\top\|^2 + 1) - \log(\Psi + 1) \right\|^2, \tag{9}$$

where $\mathcal{N}(\mathbf{W}, \mathbf{F}_i)$ suggests parameters of $\mathcal{N}$ are prescribed by $\mathbf{W}$. We use the logarithmic function $\log$ to strongly penalize the energy deviation under the same deformation and to ensure that the energy is always non-negative. Since the energy function changes smoothly with $\{e, \nu\}$, our pre-defined topological order of $\{e, \nu\}$ samples greatly eases the training. $\mathbf{W}$ can converge within only hundreds of gradient descent iterations when training uses the previous $\mathbf{W}$ for initialization. During inference, after $\mathcal{D}$ predicts $\mathbf{W}$, we apply a few extra iterations of gradient descent to fine-tune these weights, ensuring $\mathcal{N}$ fits the desired elastic energy function accurately. This two-step process ensures a smooth variation of the energy function with respect to $\{e, \nu\}$, allowing for efficient and precise generation of the network parameters.

## 4.4 Subspace encoding

If the quadratic approximation of equation 1 is exact, NeuralMTL, $\mathcal{N}$, is not needed. After obtaining $\mathbf{Q}_i$ for all voxels, we set its derivative to zero leading to:

$$\left( \frac{\mathbf{M}}{h^2} + \sum_i \mathbf{L}_i \right) \mathbf{q}_{n+1} = \mathbf{f}_q + \frac{\mathbf{M}}{h^2}(\mathbf{q}_n + h\dot{\mathbf{q}}_n) + \sum_i \mathbf{b}_i, \tag{10}$$

where $\mathbf{b}_i = \mathbf{L}_i \mathbf{q}_n - \frac{\partial E_i}{\partial \mathbf{q}}$. We refer to $\frac{\mathbf{M}}{h^2} + \sum_i \mathbf{L}_i$ as the *global matrix*, which is constant in this case. As a result, one can perform a pre-factorization converting the global matrix into lower and upper triangles to facilitate an effective solve of the linear system. However, the use of NeuralMTL alters the energy landscape nonlinearly, which makes $\mathbf{L}_i(\mathbf{q})$ dependent on the current deformed pose $\mathbf{q}$. Evaluating the system in a full implicit manner requires the information of $\nabla_{\mathbf{q}} \mathbf{L}_i$ and thus $\nabla_{\mathbf{q}} \mathcal{N}$, which is not only prohibitive but also less stable as extra training constraints need to be imposed e.g., to penalize $|\nabla \mathcal{N}|$ to prevent overfitting. To this end, we employ a lagged approach in computing $\mathbf{L}_i$ by using $\mathbf{q}$ from the most recent update.

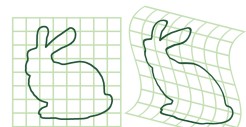

Solving the global matrix in a neural network way is challenging as all the features at vertices will become densely correlated via the matrix inverse, even the global matrix itself is sparse. To deal with this difficulty and to ensure EnasltoGen produces a physically accurate trajectory of the object, we decompose the effect of the global solve over $\mathbf{q}$ by applying multiple local operators at $\mathbf{q}_i$. Each local process works like a Laplacian operator, smoothing the rhs of equation 10 that depends on neural projection results of $\mathbf{P}_i$. Conceptually, this strategy can also be understood as finding a way to solve a linear system of the global matrix in a matrix-free manner.

Figure 2: Deforming object with the rasterization grid.

Following this inspiration, we build ElastoGen as a two-level RNN network. The outer level of RNN or RNN-1 (e.g., see figure 1) repeats local NeuralMTL adjustments over $\mathcal{G}_i$ at each voxel region and the neural projection for $\mathbf{P}_i$. The inner RNN i.e., RNN-2 tackles the global smoothing. Specifically, we repeat a local smoothing conventional kernel and approximate the global smoothing effect as the outcome of repetitive local smoothing. Each local smoothing relaxes or releases the

concentrated strain predicted by NeuralMTL $\mathcal{N}$ via expanding or shrinking its interface, which is shared with its neighboring voxels so that the local relaxation is slowly propagated over the entire object. Since a local relaxation is always applied at a voxel with eight vertices in our implementation, the corresponding network module shares the same structure for all the voxels.

A drawback of this strategy lies in the fact that it often takes a large number of RNN loops to generate a good global relaxation result. This is because local operations are more effective in processing locally concentrated strains, while object-wise global deformation can only be progressively approximated by information exchange via interface sharing across voxels. This is also a well-known limitation in numerical computation — Gauss-Seidel- or Jacobi-style iterative methods are less effective in relaxing low-frequency residual errors, which are often paired with a multigrid solver for large-scale problems.

We augment our RNN-2 with a deep encoder which extracts low-frequency strain the global matrix could generate. By encoding the input rhs of equation 10 into a low-dimension latent space of low-frequency deformations such as body-wise bending, twisting, or rotation, RNN-2 only needs to handle the remaining residual strains, which are often condensed locally. Determining the subspace encoding involves performing an SVD on the global matrix. Since our objects are rasterized, we use a rasterization grid as a general-purpose subspace. Each latent mode is visually similar to a gentle sine or cone wave e.g., see figure 2.

## 5 EXPERIMENTS

We implement ElastoGen using `Python`. Specifically, we use `PyTorch` (Imambi et al., 2021) to implement the network and a simulator for training data generation. Our hardware platform is a desktop computer equipped with an `Intel i7-12700F` CPU and an `NVIDIA 3090` GPU. Detailed statistics of the settings, models, and fitting errors are reported in table 1. *All the experiments are also available in the supplemental video*.

Table 1: **Experiments statistics.** We report detailed settings of our experiments. #**DoFs**: the average number of DOFs involved in the optimization. $\Delta t$: the size of timestep. #**R1**: the average loop count of RNN-1 for each step. #**R2**: the average number of RNN-2 loops for each timestep. **# latent**: the dimension of latent layer in the subspace encoder. **EM**: the elastic materials including Neo-Hookean (NH), StVK, and co-rotational (CR) models. **Fitting error**: the loss of NeuralMTL in equation 9. $t/$**frame**: the seconds needed for each frame.

| Scene | Grid resolution | #DoFs | #latent | $\Delta t$ | #R1 | #R2 | EM | Fitting error | $t/$frame |
|---|---|---|---|---|---|---|---|---|---|
| **ShapeNet** (Fig. 3) | $32 \times 32 \times 32$ | 5K | 36 | 0.002 | 10 | 213 | NH | $1.32 \times 10^{-4}$ | 0.08 |
| **Cantilever** (Fig. 4) | $16 \times 3 \times 3$ | 432 | 18 | 0.001 | 5 | 108 | All | $4.11 \times 10^{-4}$ | 0.01 |
| **Cantilever** (Fig. 8) | $16 \times 3 \times 3$ | 432 | 18 | 0.001 | 15 | 140 | NH | $9.67 \times 10^{-5}$ | 0.01 |
| **Lego** (Fig. 5) | $26 \times 46 \times 30$ | 11K | 54 | 0.005 | 15 | 320 | NH | $2.34 \times 10^{-4}$ | 0.44 |
| **Drums** (Fig. 5) | $28 \times 22 \times 34$ | 4K | 54 | 0.005 | 15 | 320 | CR | $7.63 \times 10^{-5}$ | 0.21 |
| **Bridge** (Fig. 7) | $66 \times 13 \times 27$ | 7K | 81 | 0.003 | 5 | 96 | StVK | $5.78 \times 10^{-4}$ | 0.92 |
| **Ship** (Fig. 7) | $53 \times 33 \times 16$ | 14K | 81 | 0.001 | 5 | 100 | NH | $2.34 \times 10^{-4}$ | 1.20 |

### 5.1 4D GENERATION FOR ANY SHAPES

ElastoGen generates 4D elastic dynamics of 3D models with any shapes. To demonstrate this, we conduct experiments on multiple models from ShapeNet (Chang et al., 2015) with arbitrary external forces and boundary conditions. Some results of ElastoGen are shown in figure 3, and more are available in the appendix. All 3D objects are rasterized with a $32 \times 32 \times 32$ grid, which also serve as our subspace encoding. Cabinets are fixed at the bottom, twisted, and then released to yield elastic oscillations. Towers and plants sway under prescribed wind fields. Airplanes are pinned at the middle. Users apply sharp dragging force at the tip of the wings, resulting in interesting and realistic dynamic effects. These results show that different boundary conditions and external forces produce plausible dynamic outcomes.

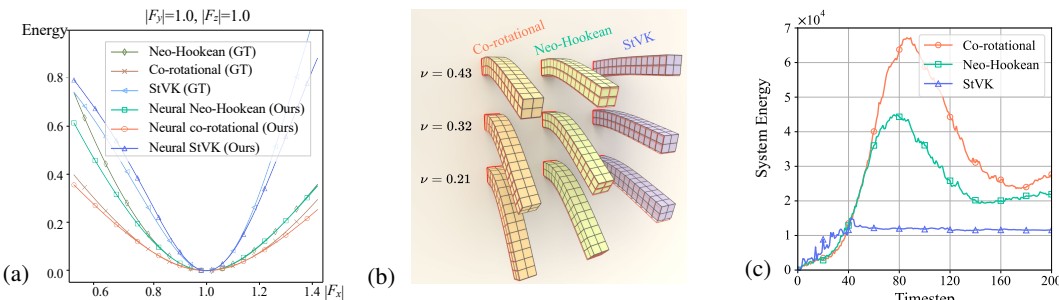

Figure 3: **ElastoGen on ShapeNet.** ElastoGen generates physically grounded 4D dynamics for objects of any geometries. To demonstrate this property, we run ElastoGen for a wide range of 3D objects in ShapeNet with different boundary conditions and external forces. This figure shows snapshots of a subset of our results including cabinets (green), towers (blue), plants (yellow), and airplanes (red). These experiments are under the rasterization resolution of $32 \times 32 \times 32$.

## 5.2 QUANTITAIVE VALIDATION OF NEURALMTL

ElastoGen replicates the behavior of real-world and complicated hyperelastic materials with different material parameters. We quantitatively compare the results generated with ElastoGen and simulated using the finite element method (FEM). We report the comparison for a standard bending test of a cantilever beam. We use ElastoGen to predict the further trajectory for three classic materials co-rotational (Brogan, 1986), Neo-Hookean (Wu et al., 2001), and StVK (Barbič & James, 2005). More general nonlinear materials, such as spline-based materials (Xu et al., 2015), are also supported. Each material is tested with three different Poisson's ratios while keeping a fixed Young's modulus (Poisson's ratio alters the material response more nonlinearly than Young's modulus). The results of ElastoGen, as shown in figure 4 (b), align well with the results obtained from the classic method of FEM. Both overlap nearly perfectly. Such superior accuracy is due to our NeuralMTL prediction. As shown in figure 4 (a), the diffusion-generated strain from NeuralMTL closely matches the ground truth (GT) with the correlation coefficient $r$ being larger than $0.98$ (calcualted as $r = \frac{\sum_{i=1}^{n}(g_i - \bar{g})(f_i - \bar{f})}{\sqrt{\sum_{i=1}^{n}(g_i - \bar{g})^2 \sum_{i=1}^{n}(f_i - \bar{f})^2}}$ for each sample point $f_i$ and $g_i$ on neural strain and the ground truth curve, and $\bar{f}$ and $\bar{g}$ are their averages). We also plot the total neural energy variation over time for those materials ($\nu = 0.32$) in figure 4 (c).

Figure 4: **Quantitative validation of NeuralMTL. (a)** Comparison between the energy computed from NerualMTL strain and the ground truth energy. **(b)** Comparison with FEM under different material parameters. The relative positional error between ElastoGen (solid bars) and ground truth (red wireframes) is less than $5\%$. **(c)** Plots of the elastic energy during the prediction.

## 5.3 VERSATILITY

ElastoGen is a general-purpose generative AI model. As long as a 3D object can be rasterized, ElastoGen deals with both explicit, e.g., as shown in figure 3, and implicit shape representations. For instance, when ElastoGen readily takes an implicit neural radiance field (NeRF) (Mildenhall et al., 2021) based model. One can conveniently employ the Poisson-disk sampling as described in Feng et al. (2023) to obtain the rasterized model. Given user-specified external forces or position

Figure 5: **ElastoGen with implicit models.** ElastoGen is compatible with both explicit and implicit models. We dense-sample the space of an implicit neural field to obtain its rasterization. Instead of running a physics simulator, ElastoGen directly yields physically accurate dynamics of the implicit model, which can be synthesized from novel camera poses. This enables a direct image-to-image generation.

constraints, ElastoGen generates its further dynamics directly via a neural network without resorting to an underlying physic simulator as used in PIE-NeRF (Feng et al., 2023). Similarly, a 3DGS (3D Gaussian splatting)-based model (Kerbl et al., 2023b) can also feed to ElastoGen for 4D generation. To show this, we report two experiments using multiple view images from the NeRF datasets as the input to ElastoGen in figure 5.

ElastoGen can benefit artists and animators by quickly producing high-quality 4D animations even for complicated models. We show such examples in figure 7 of two high-resolution objects discretized as triangle meshes. ElastoGen produces visually pleasing and physically accurate dynamics while preserving the dynamic details of the fine structures. Please refer to the supplementary video for more details. We can also inversely learn the material parameter from the video to make the generation consistent with the observation.

### 5.4 MORE COMPARISONS & ABLATION STUDY

**Comparison with ground truth.** In addition to figure 4, we further compare ElastoGen with the FEM simulation under large-scale nonlinear twisting. The comparison is based on the Neo-Hookean material. For highly nonlinear instances, the physical accuracy of ElastoGen relies on the RNN loops — more loops at both RNN-1 and RNN-2 effectively converge ElstoGen to the ground truth. Nevertheless, for general-purpose generation, fewer iterations also yield good results. The detailed experiment and error plots are reported in figure 8.

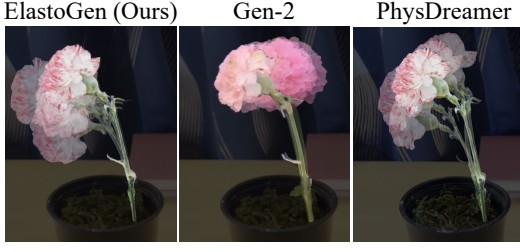

ElastoGen (Ours)    Gen-2    PhysDreamer

Figure 6: **Comparison (trajectory) between Elasto-Gen, Gen-2 (Inc.) and PhysDreamer (Zhang et al., 2024b).** We visualize the trajectory of a swinging carnation using ElastoGen, Gen-2, and PhysDreamer. Note that PhysDreamer can only produce plausible elastodynamics with tiny time steps ($\Delta t < 6.0 \times 10^{-5}$).

**Comparison with SOTA competitors.** We further compare ElastoGen with existing 4D generative models including Gen-2 (Inc.) and PhysDreamer (Zhang et al., 2024b). Elasto-Gen demonstrates superior physical accuracy and geometric consistency. Specifically, Gen-2 produces a moderate movement with very little nonlinearity like rotation and bending. In contrast, ElastoGen successfully synthesizes physically accurate large-scale motion. Gen-2 fails to maintain geometric consistency over time. Both the color of the flower and the geometry of the stem have changed using Gen-2. This is a common issue for observation-based 4D generative models, where visual correlations in training data are highly complex and challenging to be decoupled by a monolithic deep model. Note that PhysDreamer can only produce plausible elastodynamics with tiny time steps ($\Delta t < 6.0 \times 10^{-5}$) due to the underlying explicit integration, which is known to be unstable under large time steps. In contrary, ElastoGen is able to generalize on large time steps. In table 2, we present a quantitative comparison of error using the Intersection over Union (IoU) metric between ElastoGen, Gen-2 (Inc.), and PhysDreamer (Zhang et al., 2024b). The reference data is generated using Feng et al. (2023). Our method demonstrates superior accuracy in comparison to the others.

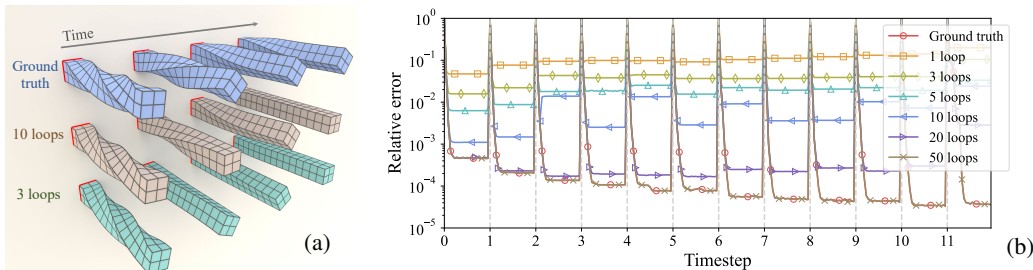

Figure 7: **ElastoGen on complex scenes.** ElastoGen seamlessly accommodates complex meshes with intricate geometries and fine structures. With subspace encoding, ElastoGen preserves both high-frequency local details and low-frequency model-wise deformations.

**Convergence study.** To quantify the impact of RNN loops and the subspace encoding on results, we compare ElastoGen predictions using different RNN loops with the ground truth, computed via solving the global matrix with a direct solver, in terms of relative error.

| ElastoGen (Ours) | Gen-2 | PhysDreamer |
|:---:|:---:|:---:|
| 94% | 64% | 75% |

Table 2: **Comparison of quantative error between ElastoGen, Gen-2 and PhysDreamer.** We compute the Intersection over Union (IoU) using reference data generated by Feng et al. (2023). Higher IoU values indicate greater accuracy.

The results and convergence plots are shown in figure 8. In this standard test, one end of the beam is fixed, and ElastoGen predicts its twisting trajectory under external forces. We note that 50 RNN loops converge ElastoGen prediction to GT. Aggressively decreasing the loop count to 20 still yields satisfactory results. In contrast, 1, 3, and 5 iterations result in noticeably stiffer dynamics. In this experiment, RNN-2 uses an 18-dimension subspace encoder to extract low-frequency resid-

uals. Without the encoding, local relaxation fails to converge.

Figure 8: **Convergence for different RNN loops.** **(a)** Comparison with FEM with different RNN loops. We note that increasing RNN loops effectively converges ElastoGen to the ground truth. However, fewer loops also give good results in general. **(b)** Relative errors for under different RNN-1 loops for each timestep. An 18-dimension subspace encoder is used to extract low-frequency residuals.

## 6 CONCLUSION

ElastoGen is a knowledge-driven deep model that embeds physical principles and numerical procedures into the network design. As a result, EasltoGen is surprisingly lightweight and compact. Each module is tailored for a well-defined computational task for minimizing the total variational energy. This design allows for decoupled training, eliminating the need for large-scale training datasets. The accuracy of ElasoGen can be easily controlled by NeuralMTL which predicts the current strain from observed numerical computations.

ElastoGen also has limitations. The current version of ElastoGen lacks the support for collisions. It becomes less efficient for thin geometry as many convolution operations is applied on empty voxels. ElastoGen may fail to converge with extremely stiff materials like a near-rigid object. In the future, we plan to keep enhancing the scope of ElastoGen e.g., by integrating dynamics for more physical phenomena such as fluids, granular materials and plasticity, adding collision support, and automating the setting of physical parameters to ultimately achieve the goal of generating real-world dynamics with mouse clicks.

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

# A APPENDIX

## A.1 SUPPLEMENTAL VIDEO

We refer the readers to the supplementary video to view the animated results for all examples.

## A.2 DIFFUSION NETWORK $\mathcal{D}$

The goal is to train a diffusion network $\mathcal{D}$ to generate the weights $\mathbf{W}$ of a corresponding NeuralMTL model $\mathcal{N}$, given the material parameters $\{e, \nu\}$. Here, $\mathbf{W}$ denotes the weights of $\mathcal{N}$, and the process is formulated as a conditional diffusion problem guided by $\{e, \nu\}$, such that $\mathbf{W} = \mathcal{D}(e, \nu)$.

To this end, we first construct a dataset consisting of 1000 paired samples of $\{e, \nu\}$ and $\mathbf{W}$, as described in § 4.3. Following the approach of Wang et al. (2024a), we utilize Latent Diffusion Models (LDM, Rombach et al. (2022)) to generate $\mathbf{W}$, as our preliminary experiments showed that directly learning $\mathbf{W}$ led to suboptimal performance. To address this, we train an autoencoder to map the network weights $\mathbf{W}$ to a 256-dimensional latent vector, in which the diffusion process is performed.

When training the diffusion model, the autoencoder remains fixed, serving solely to encode $\mathbf{W}$ into its latent representation $l$. At each diffusion timestep $t$, we introduce noise $\epsilon_t$ to $l$, resulting in $l_t = l + \epsilon_t$. The objective is to train a noise prediction model, $\epsilon_\theta(l_t, t; e, \nu)$, to estimate the noise $\epsilon_t$ at each timestep $t$, as described in § 3.2. During inference, we begin with random noise and progressively remove noise from it using the noise prediction model $\epsilon_\theta$, guided by the material parameters $e, \nu$. This iterative denoising process produces a 256-dimensional latent vector, which is subsequently passed through the decoder to generate the corresponding network weights $\mathbf{W}$.

We train the autoencoder using a learning rate of $1 \times 10^{-3}$ and the diffusion model with a learning rate of $1 \times 10^{-4}$. Both models are trained for 1000 epochs with a batch size of 64. The architecture of the autoencoder and diffusion model is detailed in table 3. Note that in diffusion process the 256-dimensional latent vector is viewed as a 1-channel $16 \times 16$ image.

| Network | Layers | #Output features | Description |
|---------|--------|------------------|-------------|
| Autoencoder | FC | 8192, 4096, 2048, 1024, 512, 256 | Encoder |
| | FC | 512, 1024, 2048, 4096, 8192, 17153 | Decoder |
| Diffusion model | Conv2D | 256, 512 | down-sample |
| | FC | 256 | Time embedding |
| | FC | 256 | $\{e, \nu\}$ embedding |
| | Conv2D | 256, 1 | up-sample |

Table 3: **Architecture of the autoencoder and diffusion model**. FC denotes the fully connected layer, and Conv2D represents the 2D convolution layer. The third column refers to the number of output features in each layer.

## A.3 CONVOLUTIONAL DEFORMATION GRADIENT

Given an input 3D object, ElastoGen rasterizes it into a set of 3D cubes or voxels. For $i$-th sub-volume inside the 3D cubes, ElastoGen uses a 3D CNN to calculate $\mathcal{G}_i$. As $\mathcal{G}_i$ has an analytic format as described in equation 8, the kernel's weights of 3D CNN can be directly computed. To be more clear, for $i$-th sub-volume containing 8 vertices, let $\mathbf{A}_i = [\mathbf{q}_1, \mathbf{q}_2, ...\mathbf{q}_8] \in \mathbb{R}^{3 \times 8}$ and $\bar{\mathbf{A}}_i = [\bar{\mathbf{q}}_1, \bar{\mathbf{q}}_2, ...\bar{\mathbf{q}}_8] \in \mathbb{R}^{3 \times 8}$ be deformed and rest-shape position of the vertices, the weights of 3D CNN can be filled with $\left[\left(\bar{\mathbf{A}}\bar{\mathbf{A}}^\top\right)^{-1}\bar{\mathbf{A}} \otimes \mathbf{I}\right] \in \mathbb{R}^{9 \times 24}$. Here, the 3D CNN has an input channel of 3, an output channel of 9 and a kernel size of $2 \times 2 \times 2$.

## A.4 GLOBAL PHASE

As stated in the main text, we need to solve the global linear system (equation 10), which requires determining $\mathbf{L}_i$ and $\mathbf{b}_i$. We abbreviate the neural strain $\mathcal{N}(\mathcal{G}_i[\mathbf{q}_i])$ as $\mathcal{N}$, rewriting equation 5, the

energy $E_i$ for element $i$ is

$$E_i = \frac{\omega_i}{2} \left\| \mathbf{F}_i \mathcal{N} - \mathbf{U}_i \mathbf{V}_i^\top \right\|_F^2. \tag{11}$$

Based on $\mathbf{L}_i \mathbf{q} - \mathbf{b}_i := \frac{\partial E_i}{\partial \mathbf{q}}$, we can obtain the expression for $\mathbf{b}_i$ and $\mathbf{L}_i$. Taking the derivative of equation 11 with respective to position $\mathbf{q}$ we obtain

$$\frac{\partial E_i}{\partial \mathbf{q}} = \omega_i \left( \mathbf{G}_i \mathcal{N} \mathcal{N}^\top \mathbf{G}_i^\top \mathbf{q} - \mathbf{U}_i \mathbf{V}_i^\top \mathcal{N} \mathbf{G}_i^\top \right), \tag{12}$$

where $\mathbf{G}_i$ is $i$-th component of $\mathbf{G}$ corresponding to element $i$. Therefore, we derive $\mathbf{L}_i$ and $\mathbf{b}_i$ as

$$\mathbf{L}_i = \omega_i \mathbf{G}_i \mathcal{N} \mathcal{N}^\top \mathbf{G}_i^\top, \quad \mathbf{b}_i = \omega_i \mathbf{U}_i \mathbf{V}_i^\top \mathcal{N} \mathbf{G}_i^\top. \tag{13}$$

As it indicates, for each voxel, we can obtain $\mathbf{b}_i$ by applying the transformation $\mathbf{G}_i^\top$ to $\mathbf{U}_i \mathbf{V}_i^\top \mathcal{N}$. For $\mathbf{G}_i$ has been trained as a convolutional kernel as described in § A.3, we can directly fetch the previously trained kernel and perform this operation.

For the linear system in equation 10, we further write it as $\mathbf{A}\mathbf{q} = \mathbf{b}$. For any diagonally dominant matrix $\mathbf{A}$, the linear system $\mathbf{A}\mathbf{q} = \mathbf{b}$ can be solved using iterative method as:

$$\mathbf{q}^{k+1} = \mathbf{D}^{-1}(\mathbf{b} - \mathbf{B}\mathbf{q}^k), \tag{14}$$

where $\mathbf{D}$ is the diagonal part of $\mathbf{A}$ and the off-diagonal part $\mathbf{B} = \mathbf{A} - \mathbf{D}$, and $\mathbf{q}^k$ is the result after $k$ loops of RNN-2. In our case, $\mathbf{A} = \frac{\mathbf{M}}{h^2} + \sum_i \mathbf{L}_i$ and $\mathbf{b} = \mathbf{f}_q + \frac{\mathbf{M}}{h^2} (\mathbf{q}_n + h\dot{\mathbf{q}}_n) + \sum_i \mathbf{b}_i$ according to equation 10. Note that we use subscript $n$ to indicate timestep and superscript $k$ as index for RNN-2 loops.

Similar to § A.3, we use a 3D CNN to implement each iteration in RNN-2. The weights of 3D CNN are filled with $-\mathbf{D}^{-1}\mathbf{B}$ and the bias is filled with $\mathbf{D}^{-1}\mathbf{b}$. The number of input channels is 78, and the number of output channels is 24, with a kernel size of 1, representing the contribution of each voxel to its 8 vertices. The iterative process is formulated as a recurrent network, i.e. RNN-2 in our paper, to solve the global system.

## A.5 BROADER IMPACT

Our model integrates computational physics knowledge into the network structure design, significantly reducing the data requirements and making both the training and network structure more lightweight. It blends the boundaries among machine learning, graphics, and computational physics, providing new perspectives for network design. Our model does not necessarily bring about any significant ethical considerations.

## A.6 MORE QUANTITATIVE VALIDATIONS

We compare the NeuralMTL strain with the ground truth under various deformed configurations. In each case, the neural energy models closely match the ground truth, demonstrating the effectiveness and expressiveness of our neural approximations for these nonlinear energy functions.

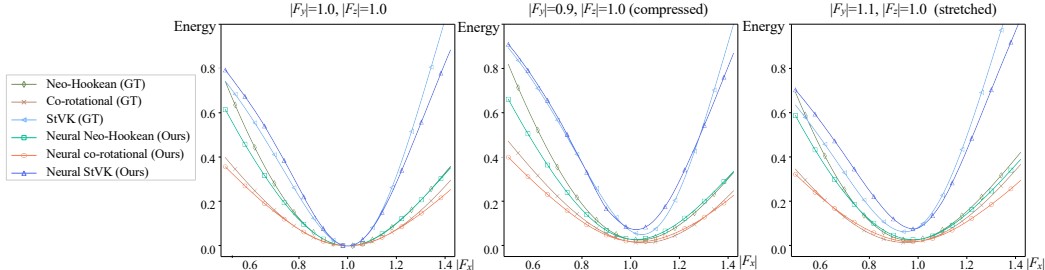

Figure 9: **More quantitative validation of NeuralMTL.** Comparison between the energy computed from NerualMTL strain and the ground truth under different configurations.

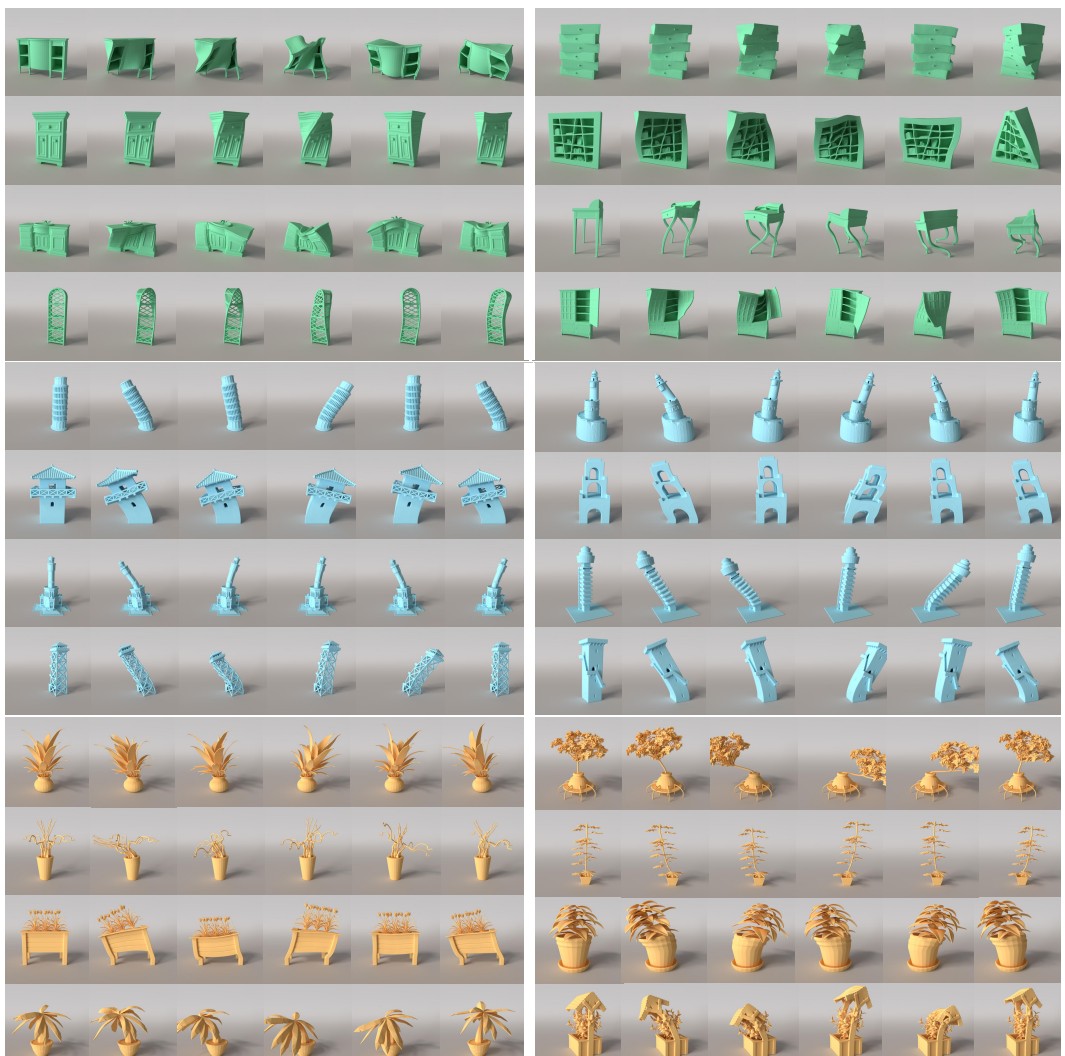

Figure 10: **Additional experiments on ShapeNet.** Here are more results of cabinets, towers, and plants.

## A.7 MORE EXPERIMENTS

We provide additional results in Fig. 10 and Fig. 11 to demonstrate the robustness of ElastoGen. For more animated results, we refer the readers to supplemental video.

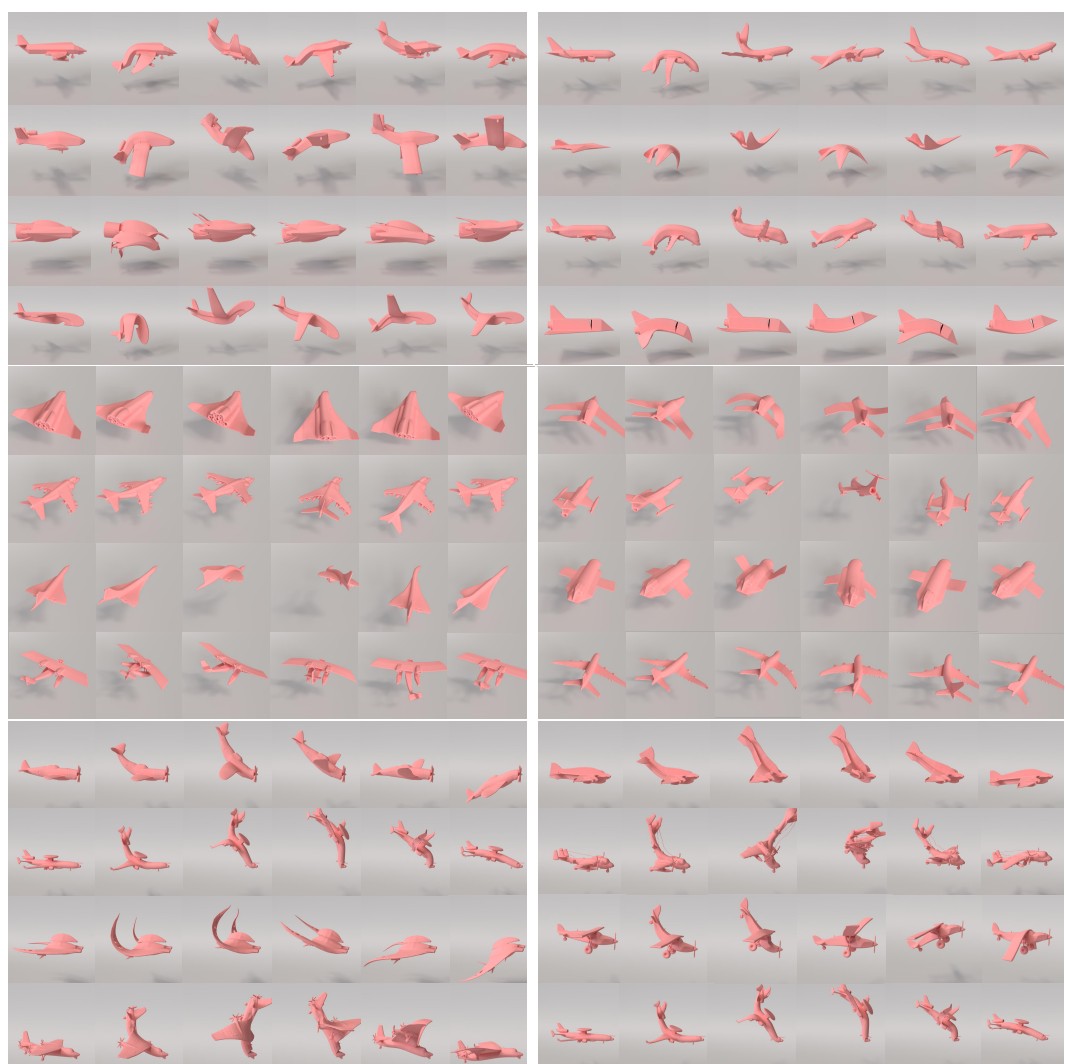

Figure 11: **Additional experiments on ShapeNet (continued).** Here are more results of airplanes with different force and boundary settings.

