# OpenReview forum: "ElastoGen: 4D Generaetive Elastodynamics"
_ICLR.cc/2025/Conference — ICLR 2025 Conference Withdrawn Submission_

### Official Review · Reviewer_ZqAh · 2024-10-31

**Soundness:** 3
**Presentation:** 2
**Contribution:** 3
**Rating:** 5
**Confidence:** 3

**Summary:**

This paper introduces ElastoGen, a knowledge-driven 4D generative model for producing accurate elastodynamics. Unlike traditional deep generative models that rely on data training, ElastoGen leverages the principles of physics by converting the nonlinear force equilibrium differential equation into iterative convolution-like operations. This approach aligns with physical models and allows for integration with other neural modules for end-to-end 4D generation. ElastoGen makes several key contributions:
1. **Compact Generative Network**: The network architecture is inspired by physical and numerical methods, reducing training complexity and creating a lightweight framework.
2. **NeuralMTL**: A NeuralMTL module adapts to various hyper-elastic materials, using conditional diffusion models to predict network parameters and reduce training costs.
3. **Nested RNN**: A two-level RNN structure, with the first level handling local strain relaxation and the second focusing on global smoothing, enhances efficiency in generating dynamics for rigid objects.

**Strengths:**

1. The paper presents an innovative approach to solving the dynamic equilibrium of 3D models by transforming the global differential operator into iterative, localized convolution-like procedures.
2. The proposed network is compact and purpose-built, free from redundant modules.
3. This method effectively handles complex real-world objects composed of hyper-elastic and isotropic elastic materials, achieving significant performance improvements over baseline methods.

**Weaknesses:**

1. The entire pipeline depends on inputting real-world material parameters, specifically Young's modulus and Poisson's ratio. However, estimating these physical material properties remains an open challenge due to the scarcity of accurate material ground-truth data, as noted in *PhysDreamer*. This limitation constrains the method’s applicability to larger-scale objects.
2. This work is specifically designed for elastic objects, though other material types, such as fluids and plastic materials, also exist. The paper notes that the network struggles to converge for near-rigid objects. Intuitively, rigid objects can be seen as extreme cases of elastic objects, and they should theoretically be easier to model. This limitation points to potential scalability challenges of the method.
3. The writing quality of the paper is subpar, with numerous spelling errors, such as "NerualMTL" on lines 216 and 250. Additionally, many sections are unclear and lack sufficient explanation, requiring clarification for better understanding. For example, on line 297, the statement "If the quadratic approximation of Equation 1 is exact, NeuralMTL, $\mathcal{N}$, is not needed" raises questions. Under what circumstances would NeuralMTL not be necessary? Furthermore, if the quadratic approximation of Equation 1 is exact in most cases, it would call into question the practical utility of this module.

It’s worth noting that I am not very familiar with this field, but I believe the weaknesses mentioned above are reasonable.

**Questions:**

1. See above.
2. On line 176, "ElastoGen automatically checks the accuracy of the prediction of both RNN loops, and outputs the final prediction of $q_{n+1}$ once the prediction error reaches the prescribed threshold." How is the prediction error evaluated during inference when ground truth dynamics are not available?
3. On line 278, "To train the model $\mathcal D$, we prepare a dataset of paired {$e, ν$} and $\bf W$." How are these $\bf W$ values collected? Are they even harder to collect compared to $e, ν$?
4. Equation 6 feels somewhat empirical—are there any ablations on alternative choices that still satisfy the symmetry and rotation-invariance requirements?
5. On line 250,  "if we choose to employ the network to learn an adjustment of $\bf{P}_i$ (which is also technically feasible), the local relaxation that predicts $\bf{P}_i$ becomes complicated, and the generation is less robust." The explanation is not clear enough for readers to understand. Why does the local relaxation that predicts $\bf{P}_i$ become complicated?

I will improve my score if the answers to the questions above are satisfactory..

---

### Official Review · Reviewer_ksPC · 2024-10-31

**Soundness:** 2
**Presentation:** 1
**Contribution:** 2
**Rating:** 3
**Confidence:** 4

**Summary:**

This works proposes ElastoGen, a hybrid model for 4D elastodynamics. It incorporates physics priors and can be embedded in a larger, differentiable deep learning model for end-to-end 4D generation.

**Strengths:**

1. This work successfully incorporates relevant physics priors in the dynamical modeling of soft materials. Incorporating stronger priors into generative models is a relevant idea.

**Weaknesses:**

1. The related work section on Generative models is extremely broad and this work is not well positioned. Ultimately, this work is compared to Zhang et al. 2024, but this is not even mentioned or discussed in the related work.
The concept of *4D* is not adequately explained. The long lists of references do not help to situate this work or understand the relevant context.

2. In terms of differentiable physics modelling, I think the authors should be aware of the related research line commonly called *differentiable physics* and discuss their method compared to those approaches:

Degrave, Jonas, et al. "A differentiable physics engine for deep learning in robotics." Frontiers in neurorobotics 13 (2019): 6.

de Avila Belbute-Peres, F., Smith, K., Allen, K., Tenenbaum, J., & Kolter, J. Z. (2018). End-to-end differentiable physics for learning and control. Advances in neural information processing systems, 31.

Hu, Y., Anderson, L., Li, T. M., Sun, Q., Carr, N., Ragan-Kelley, J., & Durand, F. (2020, January). DiffTaichi: Differentiable Programming for Physical Simulation. In International Conference on Learning Representations.

3. The experimental validation is very limited, both quantitatively and qualitatively. It is very hard to estimate the value of this work on the experimental aspect, which is very important for this kind of work.

**Questions:**

1. What are the limitations of this method compared to PhysDreamer?

2. Does the method need full access to a mesh of the objects, or can this be learned from data?

3. There are numerous typo's throughout the manuscript, even in the title of a section ('synamics') and in the acronyms ('NerualMTL').

---

### Official Review · Reviewer_HZkx · 2024-11-01

**Soundness:** 2
**Presentation:** 1
**Contribution:** 3
**Rating:** 5
**Confidence:** 3

**Summary:**

Given a 3D object and an external force, the model aims to predict the motion of the object's particle following  Lagrangian mechanics. The paper used NeuralMTL to accurately learn the potential energy concentrated at each voxel of the 3d object. Then, it solves a global matrix, which predicts the motion of voxels in the next time steps.

The method tackles a challenging problem and achieves better physical consistency than the baselines.

**Strengths:**

The paper tackles a challenging problem, offering a close physics supervision that is highly effective for generating realistic 4D simulations. This approach not only enhances the physical grounding of 4D generation but also broadens the potential applications of such simulations.

**Weaknesses:**

1. The paper is not well written. It lacks derivations and the significance of each of the operations used.

For example, in Equation 4, We have not discussed how we arrived at the shown equation. Equation 5: How do we derive the 2nd line from the first line? Equation 10: The paper does not discuss how it reached that equation. More details are asked in the Questions section.

2. The use of diffusion as a hyper-network is not well motivated. Why do we want to use a diffusion model to generate weights? Does this mean given a $(e,\nu)$, there is a distribution of different $N_i$ physically accurate for location $q_i$? This requires a clear justification.

2. Line 054: How is this a generative model? The core problem is, given a 3d object and its material properties, simulate the object's motion given an external force. So, there is a single group truth simulation based on physical laws that we want to achieve. I do not see how ElastoGen is learning a distribution here.

**Questions:**

1. Line 58, NuralMLT, this abbreviation has never been defined. Please define it when it is first used.

2. It is hard to parse this line: “ We augment ElastoGen with a low-frequency encoder, which extracts low-frequency dynamic signals so that the local relaxation only takes care of the remaining high-frequency strains.”

--> It is will good to clarify - what does **augmenting** ElasgoGen mean?  What is a "low frequency encoder"? What is a "low-frequency dynamic signal"?

3. 160- “Therefore, ElastoGen does not have redundant or purposeless network components that could potentially lead to overfitting.”—Strong claims like these should be followed up with ablation studies for different parts.

4. The input format of the object to the model must be clearly stated - in this voxel representation, polygon mesh, etc

5. 192 - what is a ‘positional feature’? is this just 3d position, i.e., (x,y,z)?

6. 192-193 - are we calling the deformation gradient “strain level feature”, because that point to the same object $F_i$
7. I am not sure how $M_i$ in line 194 is defined. It is defined as a 0 level set of $E_i$; however, $E_i$ depends on $M_i$, making this cyclic. Please clarify.

8. 190. Should it be $E_i(q_i)$?
9. From where do we get Equation 4? never described
10. line 223, what is ‘neural strain’?
11. Please explain how applying $F_i$, the deformation gradient, converts the material space strain to world space.
12. What is “SDV activation”- does it mean you perform singular value decomposition of $F_i$?
13. How do we derive the 2nd line of equation 5  from the first line? It is never explained in the paper or not in the supplementary.
14. The motivation for doing SVD is not well explained. Lines 246-248 try to explain something, but it is not clear.
15. Line 267: How do we know when NeuralMTL is not required?
16. Now, For equation 10, do we use subscript $q_n$ to denote time $n$? But in the previous section, we used the subscript $q_i$ to locate $i$. Now, it is hard to distinguish between them. Please use superscript and subscript to denote to different physical variables like time and location. Or explain if I misunderstood this.
17. Line 315: “EnlastoGen$
18. line 322:  What is ‘neural projection’?
19. Why aren’t other baselines such as PhysGaussian and PAC-NeRF not discussed?

---

### Note · Authors · 2024-11-14

I have read and agree with the venue's withdrawal policy on behalf of myself and my co-authors.